# Lightweight Detection System with Global Attention Network (GloAN) for Rice Lodging

**DOI:** 10.3390/plants12081595

**Published:** 2023-04-10

**Authors:** Gaobi Kang, Jian Wang, Fanguo Zeng, Yulin Cai, Gaoli Kang, Xuejun Yue

**Affiliations:** 1Department of Electronic Engineering, South China Agricultural University, Guangzhou 510642, China; 2Department of Computer Science, The University of Tennessee at Martin, Martin, TN 38238, USA; 3College of Computer Science and Software Engineering, Shenzhen University, Shenzhen 518060, China

**Keywords:** rice lodging, global attention network, sematic segmentation, knowledge distillation, transfer learning, unmanned aerial vehicles, smart agriculture

## Abstract

Rice lodging seriously affects rice quality and production. Traditional manual methods of detecting rice lodging are labour-intensive and can result in delayed action, leading to production loss. With the development of the Internet of Things (IoT), unmanned aerial vehicles (UAVs) provide imminent assistance for crop stress monitoring. In this paper, we proposed a novel lightweight detection system with UAVs for rice lodging. We leverage UAVs to acquire the distribution of rice growth, and then our proposed global attention network (GloAN) utilizes the acquisition to detect the lodging areas efficiently and accurately. Our methods aim to accelerate the processing of diagnosis and reduce production loss caused by lodging. The experimental results show that our GloAN can lead to a significant increase in accuracy with negligible computational costs. We further tested the generalization ability of our GloAN and the results show that the GloAN generalizes well in peers’ models (Xception, VGG, ResNet, and MobileNetV2) with knowledge distillation and obtains the optimal mean intersection over union (mIoU) of 92.85%. The experimental results show the flexibility of GloAN in rice lodging detection.

## 1. Introduction

Rice is a crucial agricultural product, and its yield and grain quality can be affected by the illness of lodging [1]. Typically, rice lodging is caused by a strong wind after heavy rain or overuse of nitrogen fertilizers [2]. When lodging occurs, it blocks the transportation of water in rice and negatively affects its nutrient absorption [1]. Severe lodging can result in up to a 50% loss in yield [2]. To mitigate the losses caused by rice lodging, accurate localization of the lodging area and timely treatment are of great importance. By tradition, the status of rice is diagnosed by agricultural experts and experienced farmers by the naked eyes, which is unsustainably labour-intensive in vast rice fields [3]. Fortunately, advancements in deep learning and remote sensing techniques provide a solution to this problem.

Remote sensing techniques, such as unmanned aerial vehicles (UAVs) and satellite imaging, have enabled quick detection of rice lodging over vast areas [4]. Over the past decade, deep learning techniques, especially convolution neural networks (CNNs), have enabled the successful applications of computer vision in various areas of human life [5]. Previous studies have demonstrated the efficacy of deep learning models and remote sensing techniques in automatically identifying rice lodging [6,7,8]. However, efficiency of the deep learning models is not an essential consideration in these works. In practice settings, the deployment of a rice lodging detection system requires consideration of computational resources. Lightweight models with satisfactory performance in detecting rice lodging are, therefore, crucial. Large-sized models tend to perform better, but their high computational cost makes them difficult to deploy. To improve the performance of small-sized models in detecting rice lodging, we adopt knowledge distillation [9] as an efficient technique to train the models in our study. By employing knowledge distillation, the small-sized models can achieve comparable performance to larger ones, which is advantageous in practical applications.

Previous research has explored the use of attention blocks to enhance the performance of CNNs in an efficient manner. These lightweight modules are able to refine feature maps and show potential for improving the performance of small-sized models with only a few additional parameters [10,11,12]. However, in many attention modules, such as those that use average-pooling and max-pooling to shrink feature maps, information may be lost, limiting the module’s ability to improve performance in practice [13]. As such, there is a need to explore alternative approaches that can more effectively leverage the information contained in feature maps.

In this study, we propose a novel attention mechanism, named the global attention network (GloAN) (shown in Figure 1), to enable efficient and accurate detection of rice lodging. The proposed GloAN takes advantage of the global receptive field and infers attention maps by simultaneously extracting spatial and channel information. Unlike the traditional attention modules, GloAN does not rely on max-pooling or average-pooling to shrink the feature maps before attention operations, which enables full utilization of the input feature maps.

In the experiments, we implemented all the segmentation models based on transfer learning [14] as it is an effective skill to improve the performance of semantic segmentation models [15,16,17]. By applying transfer learning, the implemented models can avoid training from scratch on the rice lodging dataset and can learn to distinguish subtle differences between input images before training begins. The experimental results demonstrate the effectiveness of our methods in detecting rice lodging. We summarized the main contributions of this study as follows:We propose a novel attention module, named global attention network (GloAN), which can seamlessly integrate into CNN architectures and brings notable performance gains in rice lodging detection with minimal computational cost.Our proposed methods enable accurate rice lodging detection that can be deployed with limited computational resources, which is practical for real-world applications.Our proposed methods provide a new alternative to improve the performance of semantic segmentation models in rice lodging detection. Instead of directly changing the implemented models or the backbone networks, one can improve the segmentation performance more effectively by adopting GloAN.

The remainder of this paper presents related literature in Section 2, the overview of our rice lodging detection methods in Section 3, experimental results in Section 4, discussion of our research findings in Section 5, and concluding remarks in Section 6.

## 2. Related Work

### 2.1. Remote Sensing of UAV on Rice Lodging Detection

Lodging is the most common sickness that constrains the high grain yield of rice [18,19]. Various measures have been studied to prevent rice lodging, including rice height reduction [20], applying organic fertilizer [21], precision artificial transplanting [22], and cultivation of lodging-resistance rice varieties [23,24]. However, determining the status of rice requires the assessment of experienced agricultural experts, which is time-consuming. Given that quick detection of lodging on a large scale is needed for timely treatments [6], UAV remote sensing techniques combined with deep learning can be employed to detect rice lodging.

UAV remote sensing has recently drawn broad attention from both academia and industry due to its high efficiency, flexibility, and low cost [25,26]. It provides an effective solution to monitor and detect rice lodging over vast areas [3,4,27,28]. Compared to satellite remote sensing, UAV remote sensing can use high-resolution cameras to capture high-quality images that are appropriate for lodging assessment [6,29]. By analysing UAV-captured images using deep learning techniques [30,31], rice lodging can be effectively detected.

Previous studies have shown that semantic segmentation algorithms applied to segment rice lodging produce encouraging results [6,29]. In rice lodging detection, semantic segmentation models are commonly used to localize the lodging areas and segment them from healthy rice [7]. However, most previous studies on rice lodging detection concentrate on improving model performance without reporting the computational costs of the implemented models. To achieve a reasonable trade-off between performance and computational burden, it is essential to focus on both aspects.

### 2.2. Attention Mechanism and Knowledge Distillation

For humans, attention plays a crucial part in vision understanding [32]. Usually, we perceive our surroundings by focusing on salient parts of the scene, rather than the entire scene [33]. Likewise, in deep learning architectures, attention plays a similar role and has been applied in a wide range of tasks, including image classification [10], object detection [34], and scene segmentation [35]. Instead of allocating equal weights to all positions in an input image, attention mechanisms differentiate the salient parts and unimportant parts. This means that informative features in an image are emphasized while less informative ones are suppressed [36].

SE-Net [11] has been proposed to exploit inter-channel relationships, where weights of every channel in the feature map are re-allocated. The channel attention of SE-Net is efficiently inferred only using a pooling layer and two consecutive fully connected layers. Besides channel attention, CBAM [12] also introduces spatial attention, which experimentally achieves promising performance. Typically, spatial attention plays an important role in deciding which parts of an image the model should focus on. In contrast, our proposed GloAN takes advantage of the global receptive field of the vision transformer [37] and computes channel and spatial attention simultaneously. In particular, GloAN enhances the representation power of the implemented models by modelling the inter-spatial relationship of the input features.

Models of larger size and deeper structure tend to achieve better performance [38,39]. However, compact models are preferable for deployment as they require less computational resources. Knowledge distillation was first proposed by [9] to improve the performance of a compact model by utilizing the knowledge learned by a well-trained teacher network. Based on this idea, a later work by [33] improved the student model by forcing it to mimic the feature maps produced by a powerful teacher network in the forward process. Experimental results show that this skill can yield decent performance improvement.

Knowledge distillation methods specifically designed for semantic segmentation have also been explored. To enable fast and accurate segmentation, Xie et al. [40] facilitated the learning of the student network by minimizing the probability output discrepancy and logit output discrepancy between teacher and student networks. By employing this training strategy they improved the performance of the compact segmentation model without incurring extra computational resources. A recent knowledge distillation study [41] reported to align the activation maps of teacher and student in a a channel-wise fashion. Experimental results showed that the proposed channel-wise knowledge distillation outperformed state-of-the-art distillation methods in terms of promoting training of student networks.

## 3. Materials and Methods

### 3.1. Dataset Overview

The rice lodging dataset used in this study was collected by DJI Mavic 2 Pro (DJI Innovation Technology, Shenzhen, China) in the agricultural experimental field of South China Agricultural University located in Zengcheng, Guangzhou, China ($23^{\circ }24{}^{\prime }$ N $113^{\circ }64{}^{\prime }$ E). The UAV was equipped with a Hasselblad L1D-20C digital camera (Hasselblad, Sweden, Goteborg) that captured images in 5472 × 3648 resolution and the data was collected at noon in sunny weather when the experimental field was well lit. To capture the dataset, we set up an automatic flight route for the UAV using DJI GS Pro software. The UAV flew at the height of 15 m and the speed of 5 m/s, resulting in 144 captured images that covered a paddy field of 1.5 acres. Then we used Pix4Dmapper to stitch the captured images, resulting in a single overall image with a resolution of 18,779 × 12,929. We manually annotated the healthy rice and rice lodging areas using the Labelme software (CSAIL, Boston, MA, USA). Furthermore, we carried out a field investigation to verify the ground truth of the rice paddy to ensure the accuracy of our annotations. The overview of the experimental paddy field and samples of healthy rice and rice lodging are shown in Figure 2. We highlight the rice lodging areas with a faint red colour so that they are easier to see. Note that rice lodging covers 42% of the experimental paddy field.

To created a suitable dataset for deep learning architectures, we divided the overall single image into a training set, a validation set, and a testing set at a ratio of 60, 20, and 20%, respectively. We randomly cropped 2000 images to the resolution of 224 × 224 from the overall image, covering 40% of the paddy field without overlaps. The resulting 2000 images were then equally divided into the validation and testing sets (1000 images for each set). After this, we randomly cropped 9000 images with overlaps at the same resolution from the remaining 60% of the area. During the training process, we conducted horizontal flipping, colour jittering and resizing for data augmentation. As a result, we generated a training set with 9000 images, a validation set with 1000 images, and a testing set with 1000 images.

### 3.2. Global Attention Network

The GloAN is a computationally efficient unit that can be seamlessly integrated into any CNN architecture. In detail, GloAN was built based on a mechanism called query–key–value (QKV) multiplication that helps attention-based models to selectively focus on the relevant parts of the input sequence, where query (Q), key (K), and value (V) are three matrices [42]. The QKV multiplication was originally used in natural language processing (NLP) but has been shown to be effective in computer vision tasks as well [37,43,44], where the QKV multiplication0based models are called vision transformers.

Based on the above works, we designed GloAN and applied it as an attention module for rice lodging detection. Note that GloAN is designed in a computationally efficient manner and differs from conventional vision transformers in the following distinct aspects: (1) GloAN employs a linear layer to map the input features to a low-dimensional space to reduce computational complexity. This allows GloAN to be inserted into any layer of a CNN without concern for excessive computational burden, even when the input features have a large size. (2) GloAN functions as an attention module to enhance the performance of a CNN rather than being utilized independently for result prediction or as a backbone for tasks such as object detection and semantic segmentation. (3) GloAN is a compact unit in comparison to other vision transformers which tend to be much larger in size.

As illustrated in Figure 1, given an input feature $X\in R^{H\;\times \;W\;\times \;C}$, GloAN infers an attention map $A\in R^{H\;\times \;W\;\times \;C^{\prime }}$ that captures the inter-spatial relationship of the input feature. To be specific, the input feature is firstly reshaped into a 2D vector, and subsequently forwarded to a linear layer. When propagating through the linear layer, the channel dimension of the input feature is shrunk to $C^{\prime }$ ($C^{\prime }$ is set as 64 by default and its value can vary based on the computational resources). This lessens the computational complexity, since outputs of a convolution layer generally have a huge channel dimension (e.g., 256 or 512). Secondly, we forward the reshaped feature to the second linear layer, where the output will be chunked into three matrices of the same size (denoted as $M_{1},M_{2},M_{3}$, and $\in R^{\left(H\;\times \;W\right)\;\times \;C^{\prime }}$). Note that the generated matrices have the same spatial resolution as the input feature (that is, H × W), except the spatial dimension is reshaped into one-dimension in the matrices.

Since dot-product results between vectors measure the vectors similarities [45], we perform a matrix multiplication between $M_{1}$, $M_{2}$ and $M_{3}$ to infer an inter-spatial relationship of the input feature map. Derived from $M_{1}\cdot M_{2}^{T}$, the intermediate matrix $M_{I}\in R^{\left(H\;\times \;W)\;\times \;(H\;\times \;W\right)}$ deduces the similarity between every pixel in the input features. For instance, the (H × W) elements of the first row of $M_{I}$ indicate the similarities between the first pixel and all pixels from the input feature map. Subsequently, the result matrix $M_{R}\in R^{\left(H\;\times \;W\right)\;\times \;C^{\prime }}$ is derived by the dot product of $M_{I}$ and $M_{3}$, which encodes where emphasis or suppression should be applied in the feature map, as each of its elements are inferred through global information of the input feature map, endowing it with a global receptive field. At last, a two-layer multilayer perception (MLP) is employed to recalibrate and produce the final attention map $A\in R^{H\;\times \;W\;\times \;C^{\prime }}$. Note that we do not reduce the channels of the input feature when it propagates through the MLP in contrast to what many other works do [12,43], given that the number of channels is already computationally efficient enough.

To apply attention to the input feature, we first use an average-pool to squeeze the channel of the attention map $A\in R^{H\;\times \;W\;\times \;C^{\prime }}$, which results in a refined attention map $A_{R}\in R^{H\;\times \;W\;\times \;1}$. Subsequently, the input feature is multiplied by the refined attention map $A_{R}$, where $A_{R}$ is broadcast accordingly along the channel dimension. The overall process can be summarized as:(1)$$A_{R}=AvgPool\left(Attention\left(X\right)\right),\tilde{X}=\left(A_{R}+1\right)⊗X,$$
where X denotes the input feature, $\tilde{X}$ denotes the final output of the attention transformation, and ⊗ refers to the element-wise multiplication. Figure 3 depicts the schema of the integration of GloAN into a ResNet block [39]. When applying the GloAN, an additional branch is presented in the ResNet block to infer the attention map. The attention map can perform feature recalibration by selectively emphasizing the informative parts of the feature. More variants of GloAN integrating into other CNN architectures, such as VGGNet [46], Xception [47] and MobileNet [48], can be implemented by following this scheme. Note that GloAN can be inserted into any layer of a CNN, depending on the computational budget.

### 3.3. Semantic Segmentation Models

To select a proper semantic segmentation model for rice lodging detection, we implemented three dominant algorithms, namely, U-Net [49], SegNet [17], and DeepLabv3+ [16], in this study. All of these models are composed of two parts, i.e., an encoder and a decoder. Typically, the encoder progressively downsamples the feature maps and extracts semantic information, whereas the following decoder gradually restores spatial information and enables precise object boundary recovery. In general, state-of-the-art classification, such as like VGG [46], ResNet [39], or MobileNets [48], is employed as the encoder for a semantic segmentation model, and transpose convolution or interpolation is adopted to upsample the feature maps and allow spatial resolution restoration.

Notably, unlike other semantic segmentation models we implemented, the DeepLabv3+ encoder also includes an atrous spatial pyramid pooling (ASPP) module [50], where three atrous convolutions with different atrous rates are used to capture multi-scale semantic information. More precisely, atrous convolution can effectively enlarge the receptive field of filters without learning additional parameters. Figure 4 illustrates the mechanism of atrous convolution by comparing the receptive field of atrous convolution (atrous rate set as two) and standard convolution (equivalent to the atrous rate set as one). It shows that when applying a 3 × 3 standard convolutional filter, the value of a pixel in a feature map II is determined by the 3 × 3 pixel block in the feature map I. Likewise, the values of the 3 × 3 pixel block in a feature map I are determined by the 5 × 5 pixel block from the input. As a result, the value of a pixel in a feature map II is related to the 5 × 5 pixels in the input, which means the feature map II has a receptive field of 5 × 5. In contrast, when applying a 3 × 3 atrous convolutional filter, the value of one pixel in a feature map II is determined by the 3 × 3 separated pixels in the feature map I, which are in turn related to the 7 × 7 pixel block in the input. As a result, a pixel in a feature map II represents the information contained in a 7 × 7 pixel block in the input. Thus, a larger receptive field is enabled by atrous convolution, conducive to accurate object boundaries recovery.

### 3.4. Knowledge Distillation

The key idea of knowledge distillation is to use the knowledge learned by a large, complex, well-trained teacher network to promote the training of a relatively smaller, simpler student network, so that the models in small size can achieve a better performance without learning additional parameters. In classification tasks, the Softmax function is usually used to convert the network output logits into a probability distribution. Inspired by the encouraging results achieved by channel-wise knowledge distillation [41], we employed channel-wise, instead of spatial, knowledge distillation to train the student network, where the Softmax function was applied on every single channel of outputs across the spatial dimension.

Given an input image $I\in R^{H\;\times \;W\;\times \;C}$, the output of a semantic segmentation model $Y\in R^{H\;\times \;W\;\times \;C^{\prime }}$ can be derived by Y=Model(I), where $C^{\prime }$ is the number of total classes of the dataset. In the traditional Softmax function, a well-trained teacher network tends to predict the correct class with high confidence, whereas the incorrect classes are usually assigned with probabilities close to zero, which limits their influence on model optimization. In fact, differences between incorrect classes should also be learned by the student network since they describe the similarity structure of the data. Similar to previous work in [9], a hyperparameter *T* was introduced in the Softmax function to alleviate its tendency to produce probabilities close to zero:(2)$$\theta \left(Y_{c}\right)=\frac{exp(\frac{Y_{c,i}}{T})}{\sum_{i=1}^{H\cdot W}exp\left(\frac{Y_{c,i}}{T}\right)},$$
where $Y_{c}$ refers to one channel of the output, *i* refers to every spatial location of this channel, and *H* and *W* refer to height and width of the output. With the inclusion of the hypeparameter *T*, it mitigates the tendency of the Softmax function to produce probabilities close to zero. In practice, setting the value of *T* greater than 1 leads to a softer probability distribution, which means that the differences between the probabilities assigned to the different classes are less pronounced. In our experiment, *T* was set as 4 by default as it is commonly adopted in knowledge distillation and has been found to be effective in previous studies [9,33,41].

The total loss of training the student network contains two parts, i.e., knowledge distillation (KD) loss and cross-entropy (CE) loss. To be certain, Kullback–Leibler (KL) divergence is used to measure the KD loss by evaluating the channel distribution discrepancy between the student and teacher networks, and CE loss is commonly employed in semantic segmentation tasks. The total loss of student training is formulated as [41]:(3)$$L_{Total}=L_{CE}+\alpha \cdot L_{KD},$$
(4)$$L_{kD}=\frac{T^{2}}{C}\sum_{c=1}^{C}\sum_{i=1}^{H\cdot W}\theta \left(Y_{c,i}^{T}\right)\cdot log\left[\frac{\theta (Y_{c,i}^{T})}{\theta (Y_{c,i}^{S})}\right],$$
where *C* refers to the total channel of the output, α is the weight of the KD loss, and $Y^{T}$ and $Y^{S}$ denote the output of the teacher and student, respectively. The introduction of $T^{2}$ is due to the magnitudes of the gradients are scales as $1/T^{2}$ with the introduction of *T* in the Softmax function. The training process of the student network is illustrated in Figure 5.

### 3.5. Transfer Learning Strategy

Inspired by the notable performance of transfer learning in semantic segmentation [15,16,17], we applied transfer learning in this work. In detail, all the semantic segmentation model backbones in this study were pre-trained on the ImageNet dataset [51]. ImageNet is a large-scale dataset that contains more than a million images and 1000 object categories. Models trained on the ImageNet dataset gain the ability to distinguish betwen similar objects as well as extract high-level features such as textures, corners, and edges of the input image. Since rice lodging and healthy rice share similar colours and contexts, it is important for the models to be able to recognize subtle differences. When applying transfer learning to a semantic segmentation model, it is generally recommended to use a large and diverse dataset, such as COCO [52], to pre-train the network. However, training a model on COCO requires significant computational resources. Furthermore, using a pre-trained model that is too complex for the new dataset can cause overfitting. Thus, we trained the complete network using the augmented PASCAL VOC 2012 dataset [53], which is a smaller, but well-known dataset for semantic segmentation. This dataset comprises pixel-wise annotated images of 20 object classes, with 10,528 images allocated for training and 1449 images reserved for validation purposes. In this way, we can prevent the model from overfitting and all parameters in the semantic segmentation model can avoid being trained from scratch, experimentally proven as an efficient skill to yield satisfactory results [16].

## 4. Results

### 4.1. Configuration of Experiment

To make an extensive investigation into rice lodging detection and evaluate the effectiveness of the proposed GloAN, our experiments included three sections. Firstly, we evaluated the performance of the GloAN when integrated into different prevailing semantic segmentation models for rice lodging segmentation. After verifying the best performing model, we proceeded to test the performance of the GloAN when integrated with different network backbones. Further, we compared the performance of knowledge distillation and GloAN in improving rice lodging segmentation accuracy. The effectiveness of the GloAN combined with knowledge distillation was also explored.

With the transfer learning strategy described in Section 3.5, the implemented models quickly converged within 40 epochs when trained on our rice lodging dataset. Since the backbone is the key unit for a semantic segmentation model to extract semantic information, we froze the backbone parameters in the first 20 training epochs and merely fine-tuned the decoder parameters. In the latter 20 epochs, we unfroze the backbone and fine-tuned the parameters of the entire network. Notice that all the parameters in our models were initialized with [54] default PyTorch initialization methods except those in the pre-trained backbones.

In our experiment, the initial learning rate started from $5e^{-4}$ and was multiplied by 0.92 every epoch. Considering computational resources, we set the batch size to 16 when the backbone was frozen and set the batch size to 4 when we unfroze the backbone. For knowledge distillation, we set the hyperparameter *T* to 4 and the hyperparameter α to 3, as shown in Equations (Equation 2) and (Equation 3). Note that we did not tune the hyperparameters very hard in our experiments since we aimed to present impartial comparisons among the different methods and produce unbiased results. The evaluation metrics in our experiments included mean intersection over union (mIoU), network parameters, and Giga floating-point of operations (GFLOPs).

### 4.2. GloAN Integrated into Semantic Segmentation Models

In this experiment, we evaluated the performance of various semantic segmentation architectures, including SegNet [17], U-Net [49], and DeepLabV3+ [16] in detecting rice lodging. We then integrated GloAN into each of these architectures and evaluated its performance. The effectiveness of transfer learning was also evaluated. We report the experimental results in Table 1. Note that all the segmentation architectures adopted the same backbone, VGG-16 [46], for the sake of impartiality.

Firstly, the experimental results show that transfer learning is a fundamental skill enhancing the performance of the segmentation models in detecting rice lodging. Secondly, all the architectures integrated with GloAN outperformed the original models by a clear margin, demonstrating that GloAN generalizes well in detecting rice lodging. Specifically, the integration of GloAN led to a 3.80% increase in mIoU for U-Net and SegNet. Furthermore, the GloAN was efficient in terms of computational complexity and model size, enabling an improvement in performance with only a slight increase in computational overhead. The experimental results highlight the great potential for the GloAN to be applied in real-world application of detecting rice lodging. Our following experiments were conducted based on DeepLabV3+ as it achieved the best performance in our evaluation.

### 4.3. GloAN Integrated into Different Backbones

Based on DeepLabV3+, we report the performance of GloAN integrated into different well-established backbones, including VGG [46], Xception [47], ResNet [39], and MobileNetV2 [48]. Table 2 summarizes the experimental results, which show that all the models incorporating GloAN yield significant improvement in performance in detecting rice lodging, demonstrating that GloAN is not only effective in different segmentation models but also performs well with different backbones, indicating its potential for practical applications.

In Table 2, we first present the performance of a large-sized backbone, ResNext-101 [38], in rice lodging detection. It shows that ResNext-101 is substantially larger in size compared to the other employed backbones, and therefore achieved the best mIoU in segmenting rice lodging (without GloAN). In practice, however, deploying such a large model would be challenging, hence smaller backbones are preferred. By inserting GloAN into these relatively smaller backbones, GloAN improved the accuracy of all the models with only a few additional parameters. In particular, GloAN led to a 3.56% mIoU improvement when integrated into the Xception backbone. Notably, smaller models even outperformed ResNext-101 in detecting rice lodging when incorporatedv with GloAN. For example, with five times fewer parameters, the VGG-16 integrated with GloAN outperformed ResNext-101 backbone in detecting rice lodging. In addition, GloAN improved the performance of ResNet-18 by 3.23%. With only half the computational complexity and model size as Xception, ResNet-18 integrated with GloAN outperformed Xception in detecting rice lodging. Making use of the GloAN, we can choose models of appropriate sizes to detect rice lodging based on our computational resources. In particular, When using MobileNetV2 integrated with GloAN as the backbone, DeepLabV3+ only occupied 5.94 MB of the memory, highly convenient for deployment even in mobile devices [48]. Figure 6 presents the confusion matrix of DeepLabV3+ with VGG-16 incorporating GloAN as the backbone in segmenting rice lodging.

Furthermore, we tested the performance of knowledge distillation in improving the performance of small-sized models. Table 2 reports the results of channel-wise knowledge distillation when taking ResNext-101 as the teacher model and ResNet-18 as the student model. The table shows that the model yielded more than 2% mIoU improvement in rice lodging detection without imposing any increase in model size or computational complexity. This demonstrates the great potential of knowledge distillation in practical applications where computational resources are limited. Figure 7 presents the effectiveness of knowledge distillation in improving the performance of small network. Note that the backbones were frozen in the first 20 training epochs.

### 4.4. GloAN Combined with Knowledge Distillation

Since both GloAN and knowledge distillation benefit model performance in rice lodging detection. We explored the potential to enhance rice lodging detection accuracy by combining GloAN with knowledge distillation. We conducted experiments to evaluate this strategy and the results are reported in Table 3.

As shown the table, GloAN yielded a 3.23% mIoU improvement and outperformed knowledge distillation in promoting model performance on rice lodging detection. More importantly, the performance of GloAN was further improved by knowledge distillation, leading to a 4.43% mIoU gain. This combination of techniques resulted in a more efficient module for enhancing model performance in rice lodging detection. Based on the ResNet-18 backbone, we visualize the rice lodging detection results in Figure 8, and compare the performance of DeepLabV3+ with and without integrating GloAN. DeepLabV3+ with the ResNet-18 backbone produced detection results where rice lodging areas were roughly localized. In contrast, DeepLabV3+ incorporating GloAN (trained with knowledge distillation) localized the rice lodging areas much more precisely.

## 5. Discussion

In this study, we aimed to achieve accurate and efficient rice lodging detection with the help of UAV remote sensing and deep learning techniques. We collected rice lodging data using a UAV equipped with a high-resolution camera. The captured images were stitched using Pix4Dmapper, and we generated the dataset by cropping the overall image to a resolution of 224 × 224. We divided the dataset into a training set with 9000 images, a validation set with 1000 images, and a testing set with 1000 images.

We employed prevailing semantic segmentation models to identify rice lodging based on transfer learning and knowledge distillation. We proposed a lightweight attention module, named GloAN, to detect rice lodging. GloAN was integrated into different semantic segmentation models and backbone networks, resulting in significant performance gains.

In our experiments, we evaluated the performance of GloAN in enhancing the accuracy of different semantic segmentation models in detecting rice lodging. The results showed that GloAN improved the performance of all the semantic segmentation models by a significant margin with minimal additional computational costs. We also tested the performance of GloAN by inserting it into different backbone networks and observed how the rice lodging detection accuracy varied. GloAN showed excellent performance and proved its generalization ability. In particular, GloAN improved the performance of Xception by 3.56% with a negligible number of parameters. Table 3 shows that GloAN outperformed knowledge distillation in promoting the accuracy of small models and showed the potential of combining GloAN and knowledge distillation to produce even better performance improvements.

Detecting crop lodging with UAV remote sensing and deep learning techniques has been widely investigated in recent years. Previous studies [8,29] have used deep learning models to extract rice lodging based on both digital and multispectral images and obtained decent experimental results. These studies may provide us with the idea of using multispectral cameras to collect rice lodging datasets it is difficult to accurately detect rice lodging with digital images. Zhang et al. [7] employed DeepLabV3+ to extract wheat lodging at different wheat growth stages and found that DeepLabV3+ outperformed U-Net in detecting wheat lodging, consistent with our findings in this study. Yang et al. [6] used FCN and SegNet to detect rice lodging based on visible images and achieved a top F1-score of 0.80. However, these previous studies did not report the computational cost of their models or comprehensively explore the potential of deep learning techniques to improve the results, a major shortcoming of these studies in our opinion. Our study attempted to addresses this gap by proposing GloAN and report the computational costs of each implemented model in our experiment, which we believe facilitates a more in-depth analysis of the experimental results.

## 6. Conclusions

In this paper, we introduced a novel approach to improving the performance of small models to detect rice lodging. We aimed to bring considerable performance gains at a minimal computational cost. To achieve this, we proposed the GloAN that effectively refines the intermediate feature maps by taking advantage of the global receptive field. Specifically, the GloAN can fully utilize the information contained in the input features by inferring the inter-spatial and inter-channel relationship simultaneously. Our extensive experiments demonstrate that the proposed GloAN can bring significant performance improvements in detecting rice lodging. In addition, we observed that our GloAN performed well combined with knowledge distillation, which further promoted the performance of deep learning architectures. Integrating GloAN yielded substantial performance improvements with a negligible number of additional parameters, enabling small models to achieve satisfactory performances in detecting rice lodging. In contrast, directly applying larger models or complex backbones, traditionally performed to improve rice lodging detection accuracy, becomes inefficient. We believe our study can provide insights into how to efficiently detect rice lodging and serve as technical support for future research on rice lodging detection.

## Figures and Tables

**Figure 1 plants-12-01595-f001:**
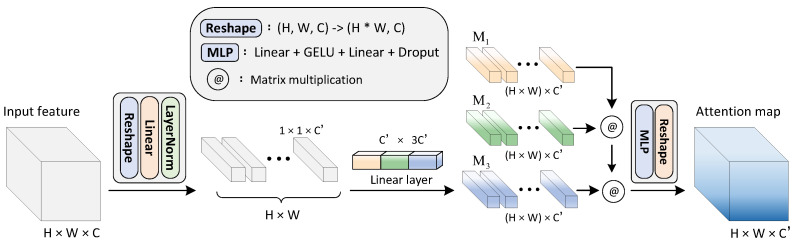
Diagram of the global attention network.

**Figure 2 plants-12-01595-f002:**
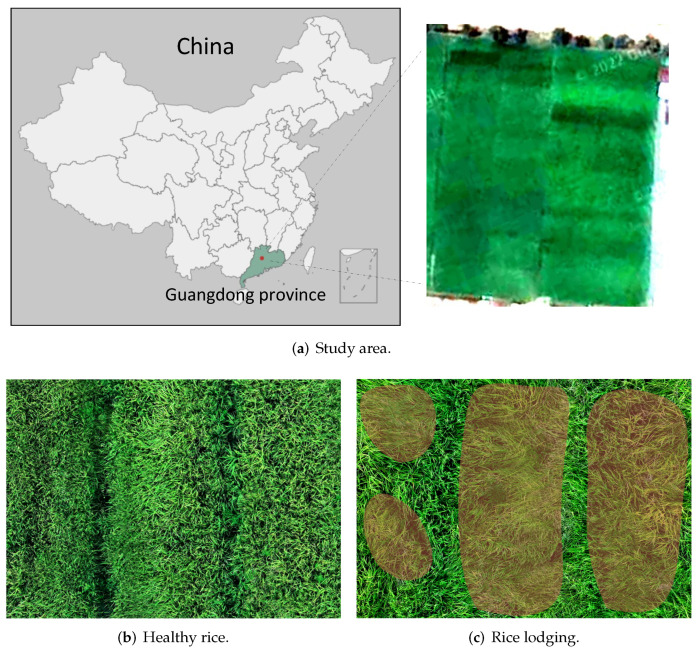
Dataset overview. The lodging areas in the rice field are highlighted in red.

**Figure 3 plants-12-01595-f003:**
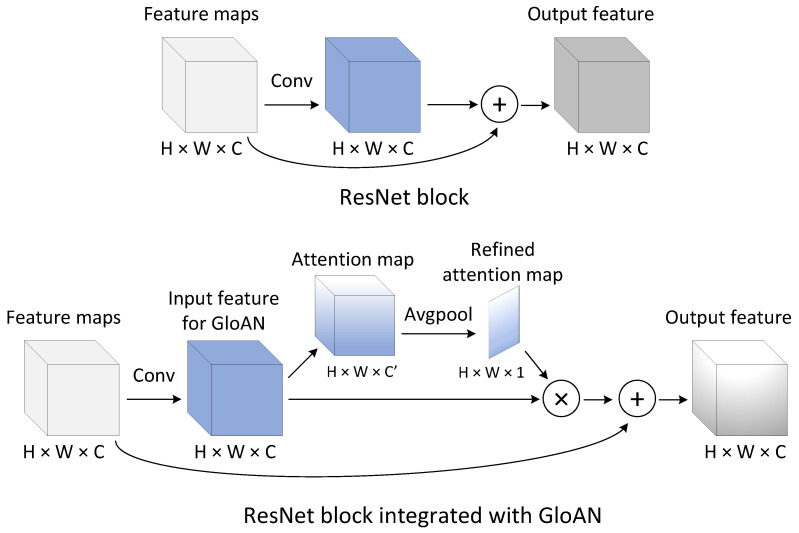
The schema of GloAN integrated into a ResNet block.

**Figure 4 plants-12-01595-f004:**
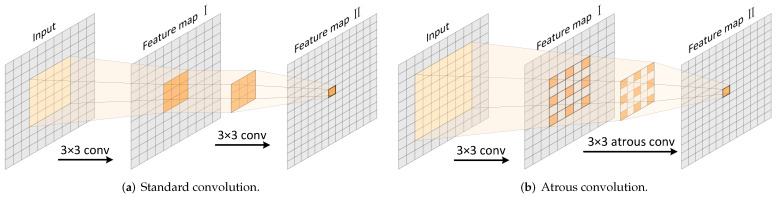
Standard convolution and atrous convolution. The receptive field of a feature map II is enlarged by implementing atrous convolution.

**Figure 5 plants-12-01595-f005:**
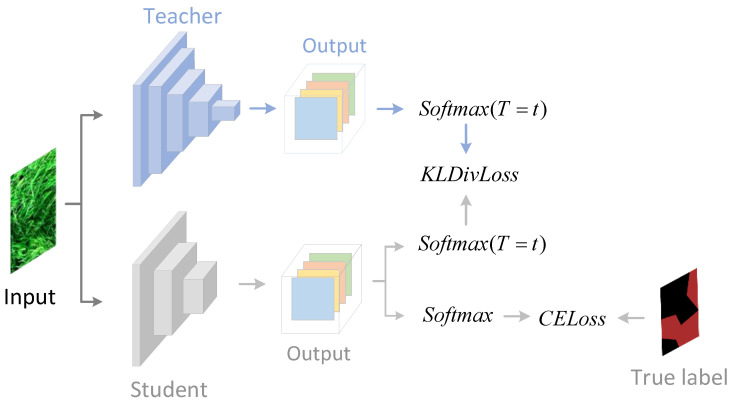
Training the student network with knowledge distillation.

**Figure 6 plants-12-01595-f006:**
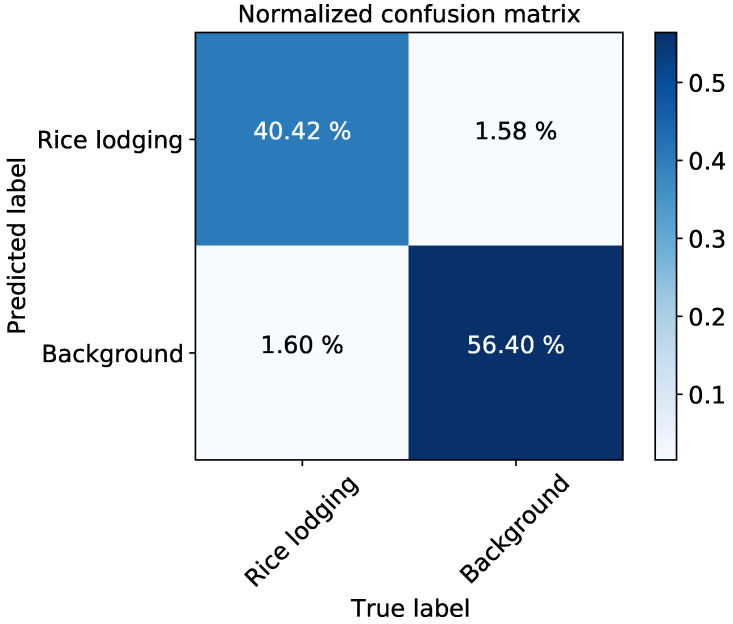
Predicting rice lodging using DeepLabV3+ with VGG-16 + GloAN as the backbone.

**Figure 7 plants-12-01595-f007:**
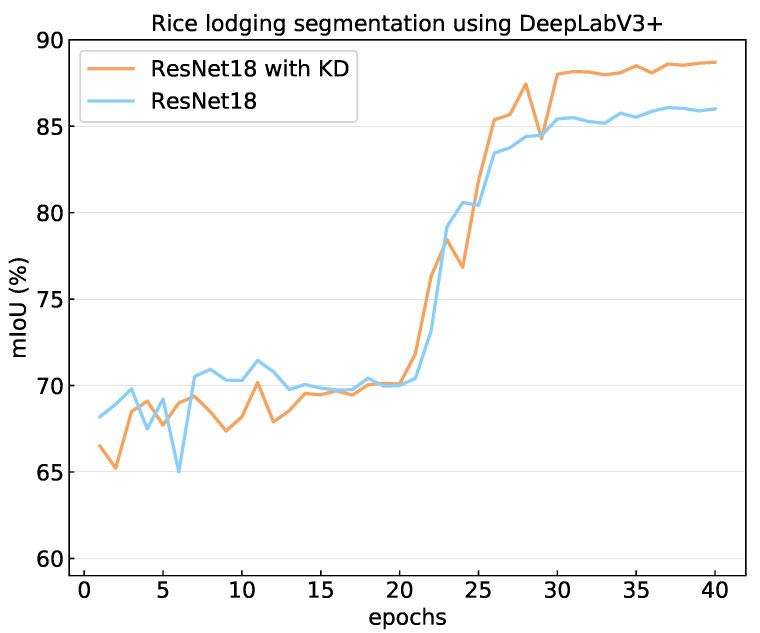
Performance of channel-wise knowledge distillation. Note that ResNet18 is the backbone network of DeepLabV3+.

**Figure 8 plants-12-01595-f008:**
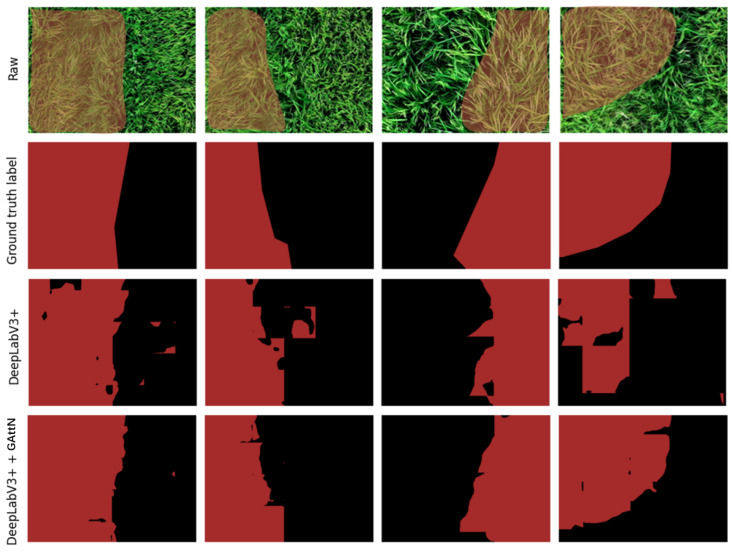
Samples of rice lodging detection.

**Table 1 plants-12-01595-t001:** Segmentation results on the rice lodging dataset.

Model	Parameters	GFLOPs	mIoU
SegNet * [17]	25.63M	135.36	**52.29%**
U-Net * [49]	28.04M	196.89	56.92%
DeepLabV3+ * [16]	20.16M	105.98	30.41%
SegNet [17]	25.63M	135.36	83.91%
U-Net [49]	28.04M	196.89	86.61%
DeepLabV3+ [16]	20.16M	105.98	**91.16%**
SegNet [17] + GloAN	25.82M	136.18	87.71%
U-Net [49] + GloAN	28.23M	197.72	90.41%
DeepLabV3+ [16] + GloAN	20.35M	106.70	**92.85%**

* trained without transfer learning.

**Table 2 plants-12-01595-t002:** Rice lodging segmentation based on DeepLabV3+.

Backbone	Parameters	GFLOPs	mIoU
ResNext-101 [38]	**105.63M**	**111.36**	**92.66%**
Xception [47]	37.64M	65.06	87.34%
VGG-16 [46]	20.16M	105.98	**91.16%**
ResNet-18 [39]	17.12M	31.77	86.07%
MobileNetV2 [48]	5.81M	26.39	83.92%
Xception [47] + GloAN	38.61M	66.67	90.90%
VGG-16 [46] + GloAN	20.35M	106.70	**92.85**%
ResNet-18 [39] + GloAN	17.50M	33.09	89.30%
MobileNetV2 [48] + GloAN	5.94M	27.00	85.66%
ResNet-18 *	17.12M	31.77	**88.70%**

* trained with knowledge distillation [41].

**Table 3 plants-12-01595-t003:** Rice lodging segmentation based on DeepLabv3+. KD denotes knowledge distillation.

Backbone	GloAN	KD [41]	Parameters	mIoU
ResNet-18 [39]			17.12M	86.07%
ResNet-18 [39]		✓	17.12M	88.70%
ResNet-18 [39]	✓		17.50M	89.30%
ResNet-18 [39]	✓	✓	17.50M	**90.50%**

## Data Availability

The dataset and source code used in this study have been open sourced and are publicly available at https://github.com/Stephenkgb/GloAN-and-rice-lodging-dataset.

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
