# Peer review of "Lightweight Detection System with Global Attention Network (GloAN) for Rice Lodging"

_plants, 2023, doi:10.3390/plants12081595_

Round 1

Reviewer 1 Report

This paper used UAV to acquire the distribution of rice growth, and proposed Global Attention Network (GANet) to detect the lodging areas efficiently and accurately. The methodology is round and the experimental results are acceptable. Some specific comments are provided as follows.

1. The authors labeled the rice lodging area using Labelme software. How to determine the rice area on the UAV images? By whom?

2. The authors employ channel-wise, instead of spatial, knowledge distillation to train the student network, where Softmax is applied on every single channel of outputs across spatial dimension. A hyper-parameter T is introduced in the Softmax function to alleviate its tendency to produce probabilities close to zero. Why and how to set T as 4 in knowledge distillation implementation? How does T affect the probability contribution?

3. The total loss of the training of student network contains two parts, i.e., knowledge distillation (KD) loss and cross-entropy (CE) loss, as Equations (3) and (4). are these two equations formulated by the authors? Otherwise, please provide the proper reference. 

4. "As shown in Figure. 6, after applying transfer learning the VGG-16 model can effectively extract high-level features like textures, corners and edges of the input image, and distinguishes two dogs from background." Why not three dogs instead of two dogs? Consider to remove this paragraph due to its irrelevant to this work of rice lodging identification.

5. Refence style should be consistent. Authors’ names are cited wrong in Refence 3. The paper tittle is wrong in Refence 4 by copying it from Reference 3. Please check all references.

Author Response

Dear Reviewer,

Thank you for taking the time to review our paper. We appreciate your valuable feedback and suggestions for improving our work. Please find attached our point-by-point response to your comments.

We hope that our revisions have addressed all of your concerns, and we look forward to hearing from you soon. 

Sinerely,

Gaobi Kang

Reviewer 2 Report

This is a research project on developing lightweight deep learning with GANet to detect rice lodging. The research sounds interesting and could be valuable. However, there are a few things remained to be clear:

1. Based on the authorship of the manuscript, there is no agriculturally related scientists involved for this agricultural application research of technology. This may cause misunderstanding and reduced confidence in the research and application communities.

2. Is GANet originally proposed in this manuscript? I note that in the description of GANet no any citation included. Seems everything original here. Be careful when you say "propose". I see it is only a modification of the classic architecture. Be clear what your is and what others are.

3.  Please check all parts of the manuscript to make sure the clarity of the writing. For example, lines 165 - 173, very confusing between the numbers 0.6. 0.2, 0.2/2000/40%, 60%/9000, 1000, 1000???

Author Response

(The authors gave the same response as above.)

Round 2

Reviewer 2 Report

Appreciate the authors' effort and all my major concerns have been well addressed. Thanks!

Author Response

-